# IL-10 Enhances the Inhibitory Effect of Adipose-Derived Stromal Cells on Insulin Resistance/Liver Gluconeogenesis by Treg Cell Induction

**DOI:** 10.3390/ijms25158088

**Published:** 2024-07-25

**Authors:** Hsiao-Chi Lai, Pei-Hsuan Chen, Chia-Hua Tang, Lee-Wei Chen

**Affiliations:** 1Department of Surgery, Kaohsiung Veterans General Hospital, Kaohsiung 813, Taiwan; kkkkkkoc@hotmail.com (H.-C.L.); hsuan901@yahoo.com.tw (P.-H.C.); chtang@vghks.gov.tw (C.-H.T.); 2Institute of Emergency and Critical Care Medicine, National Yang Ming Chiao Tung University, Taipei 112, Taiwan; 3Department of Biological Sciences, National Sun Yat-sen University, Kaohsiung 804, Taiwan

**Keywords:** mesenchymal stem cells, STAT3, adiponectin, DPP4, Foxp3

## Abstract

The modulation of cellular phenotypes within adipose tissue provides a potential means for therapeutic intervention for diabetes. Endogenous interleukin-10 (IL-10) protects against diet-induced insulin resistance. We examined the effects and mechanisms of action of IL-10-treated adipose-derived stromal cells on diabetes-induced insulin resistance and liver gluconeogenesis. We harvested stromal vascular fractions (SVFs) from the adipose tissue of diabetic (*Lepr^db/db^*) mice and treated them with IL-10 in vitro. SVFs treated with 10 or 100 ng of IL-10 were injected into the inguinal adipose tissue of *Lepr^db/db^* mice. IL-10 treatment suppressed the mRNA expression of IL-6, IL-33, CCL2, TNF-α, and IL-1β. Additionally, it suppressed the protein expression of IL-6, pmTOR, pJNK, and pNF-κB but enhanced Foxp3 mRNA expression in SVFs from diabetic mice. Meanwhile, IL-10 treatment repressed CCL2 and PDGFRα expression in adipose tissue macrophages (ATMs) and IL-6 expression in non-ATMs but increased the Foxp3 and IL-10 mRNA expression of ATMs from diabetic mice. Injection of IL-10-treated SVFs decreased the IL-6, IL-33, CCL2, IL-1β, and CCL2 but enhanced the Foxp3 and IL-10 mRNA expression of adipose tissue from *Lepr^db/db^* mice. Furthermore, injection of IL-10-treated SVFs increased CD4^+^ regulatory T cells (Tregs) in SVFs and adipose IL-10 levels and suppressed plasma adiponectin levels and DPP4 activity in diabetic mice. Injection of IL-10-treated SVFs decreased hepatic G6PC and PCK1 mRNA expression and increased Akt activation, STAT3 phosphorylation in the liver, and glucose tolerance in diabetic mice. Our data suggest that IL-10 treatment decreases inflammation in adipose SVFs of diabetic mice. Injection of IL-10-treated SVFs into the adipose tissue decreased diabetes-induced gluconeogenesis gene expression, DPP4 activity, and insulin resistance by enhancing Treg cells in diabetic mice. These data suggest that IL-10-treated adipose stromal vascular cells could be a promising therapeutic strategy for diabetes mellitus.

## 1. Introduction

Innate (macrophages, neutrophils, eosinophils, mast cells) and adaptive (regulatory T cells, type 1 helper T cells, CD8^+^ T cells, and B cells) immune cells play key roles in adipose tissue inflammation and metabolic dysregulation [1]. Tregs, characterized by the expression of the transcription factor forkhead box p3 (Foxp3), provide critical defense against abnormal immune responses such as inflammation, infection, and allergy [2]. A relatively high number of Tregs (40% of T cells) are present in adipose tissue compared to other lymphoid tissues in mice [2]. The number of Tregs in epididymal fat is significantly reduced in obese animals, which is associated with the development of insulin resistance [3]. However, the regulatory mechanisms of Tregs in adipose tissue inflammation in diabetes have not yet been fully elucidated.

Inflammation of white adipose tissue (WAT) has been observed in the adipose tissue of obese human and animal models, during which the adipose tissue is extensively infiltrated with immune cells, such as macrophages, dendritic cells, and lymphocytes [1]. Although inflammatory responses can be stimulated by various mechanisms, including hyperglycemia-induced cell death, which increases the aggregation of macrophages in the kidney [4], the mechanisms of interleukin-10 (IL-10) in the adipose tissue under conditions of chronic inflammation and related complications in diabetes have not been well characterized.

IL-10 is a pleiotropic cytokine produced by different cell types. Myeloid dendritic cells and macrophages express IL-10 upon activation of MyD88- and TRIF-dependent TLR pathways such as TLR4 and TLR3, via stimulation with LPS and dsDNA, respectively [5]. IL-10 is an immunoregulatory cytokine that plays an important role in the modulation of inflammation [6]. Studies have shown that endogenous IL-10 protects the liver against diet-induced insulin resistance [7]. Furthermore, the inhibition of IL-10 leads to the increased expression of inflammatory cytokines, worsened insulin signaling, and activation of glucogenic and lipogenic pathways [7]. Moreover, ATM-derived IL-10, induced by insulin and lipopolysaccharides, suppresses hepatic glucose production in cooperation with insulin [8]. The protective effects of IL-10 have also been reported in humans, as suggested by the strong correlation between increased IL-10 expression and the attenuation of insulin-dependent diabetes mellitus (DM) [9]. The mechanisms proposed for the regulation of T1D by IL-10 involve increases in regulatory T cell (Treg) frequencies and Th2-type cytokine (IL-4 and IL-10) levels, and suppression of IL-2 and IFN-γ cytokines [10]. Despite the immunosuppressive effects of IL-10, its role in modulating disease progression in DM remains controversial.

Adipose tissues exhibit pronounced cellular and metabolic plasticity, and the modulation of cellular phenotypes within adipose tissue provides a plausible therapeutic intervention for diabetes [11]. Although stromal stem cells are emerging as a promising source for allogenic cell therapy, the therapeutic use of mesenchymal stem cells (MSCs) in DM clinical trials is highly controversial [12,13]. We examined the effects and mechanism of IL-10-treated adipose stromal cells on diabetes-induced gluconeogenesis and insulin resistance. We hypothesized that the injection of IL-10-treated adipose stromal cells could reverse diabetes-induced gluconeogenesis and insulin resistance by inducing Treg cells within the adipose tissue. In the future, the injection of IL-10-treated adipose stromal cells could be a novel and promising cell therapy strategy for alleviating insulin resistance, gluconeogenesis, and DPP4 activity in DM.

## 2. Results

### 2.1. IL-10 Suppresses IL-6, IL-33, CCL2, TNF-α, and IL-1β Cytokines but Enhances Foxp3 mRNA Expression in Stromal Vascular Fractions (SVFs)

To examine the effect of IL-10 on inflammatory cytokine and Foxp3 mRNA expression in SVFs of the inguinal adipose tissue from type 2 diabetic (*Lepr^db/db^*) mice, SVFs (2 × 10^7^ cells) purified from the adipose tissue of *Lepr^db/db^* mice were treated with PBS, 10 ng of IL-10, or 100 ng of IL-10 for 3.5 h followed by qPCR analysis of IL-6, IL-33, Foxp3, CCL2, IL-1β, and TNF-α cytokine mRNA expression. First, we compared cytokine expression in SVFs from diabetic (*Lepr^db/db^*) and non-diabetic (*Lepr^+/+^*) mice. SVFs from *Lepr^db/db^* mice exhibited a significant increase in IL-6 and IL-33 mRNA expression compared to SVFs from *Lepr^+/+^* mice (Figure 1). Treatment of SVFs from *Lepr^db/db^* mice with 10 ng of IL-10 resulted in a significant decrease in IL-6, IL-33, CCL2, TNF-α, and IL-1β mRNA expression and a significant increase in Foxp3 mRNA expression compared with those treated with PBS (Figure 1). Moreover, treatment of SVFs of the adipose tissue from *Lepr^db/db^* mice with 100 ng of IL-10 exhibited a significant decrease in the mRNA expression of IL-6, IL-33, and IL-1β and a significant increase in Foxp3 mRNA expression compared with those treated with PBS. Collectively, these results suggest that non-diabetic SVFs exhibited decreased expression of IL-33 and IL-6 compared with diabetic SVFs. Most importantly, our results reveal that IL-10 treatment is capable of suppressing IL-6, IL-33, CCL2, TNF-α, and IL-1β cytokines but enhancing Foxp3 cytokine expression in diabetic SVFs.

### 2.2. IL-10 Suppresses IL-6, pmTOR, pJNK, and pNF-κB Protein Expression in SVFs

To examine the effects of IL-10 treatment on IL-6, pmTOR, pJNK, and pNF-κB protein expression in SVFs of adipose tissue from type 2 diabetic (*Lepr^db/db^*) mice, SVFs harvested from the adipose tissue of *Lepr^db/db^* mice were treated with PBS, 10 ng of IL-10, or 100 ng of IL-10 for 3.5 h followed by Western blot analysis of IL-6, pmTOR, pJNK, and pNF-κB protein expression. First, we compared cytokine expression in SVFs from diabetic (*Lepr^db/db^*) and non-diabetic (*Lepr^+/+^*) mice. SVFs from *Lepr^db/db^* mice exhibited a significant increase in IL-6 and pmTOR protein expression compared with those from *Lepr^+/+^* mice (Figure 2A). Treatment of SVFs from *Lepr^db/db^* mice with 10 or 100 ng IL-10 resulted in a significant decrease in IL-6, pmTOR, pJNK, and pNF-κB protein levels compared to those treated with PBS (Figure 2A and Appendix A). Collectively, these results suggest that diabetic SVFs exhibit increased expression of IL-6 and pmTOR compared to non-diabetic SVFs. Most importantly, IL-10 treatment suppresses IL-6, pmTOR, p-JNK, and pNF-κB protein expression in diabetic SVFs.

### 2.3. IL-10 Treatment Decreases CCL2 and PDGFRα but Increases IL-10 and Foxp3 Expression in Adipose Tissue Macrophages (ATMs)

To examine whether IL-10 treatment decreases CCL2 and increases Foxp3 as well as IL-10 cytokine expression in ATMs of SVFs from diabetic mice, SVFs harvested from the adipose tissue of *Lepr^db/db^* mice were treated with PBS, 10 ng of IL-10, or 100 ng of IL-10 for 3.5 h. ATMs and non-ATMs (NATMs) were purified from the SVFs followed by qPCR analysis to assess the mRNA expression levels of CCL2, PDGFRα, Foxp3, IL-6, and IL-10. Our results showed that treatment with 10 or 100 ng of IL-10 significantly decreased CCL2 and PDGFRα mRNA expression but increased Foxp3 and IL-10 mRNA expression in ATMs of SVFs from the adipose tissue of *Lepr^db/db^* mice compared with that in the control PBS group (Figure 2B). In addition, we observed that treatment with 10 or 100 ng of IL-10 significantly decreased IL-6 mRNA expression in the NATM group compared to that in the control PBS group. Collectively, our results demonstrate that IL-10 treatment represses CCL2, PDGFRα expression of ATMs, and IL-6 expression of NATMs. On the other hand, there is an increase in Foxp3 and IL-10 expression in ATMs of diabetic mice.

### 2.4. Injection of IL-10-Treated SVFs Decreases Expression Levels of IL-6, IL-33, IL-1β, and CCL2 but Enhances Foxp3 and Il-10 Expression in the Adipose Tissue

Next, we investigated whether injection of IL-10-treated SVFs would have effects on IL-6, IL-33, CCL2, IL-1β, TNF-α, IL-10, and iNOS cytokine mRNA expression in SVFs of adipose tissue from *Lepr^db/db^* mice. SVFs treated with 10 or 100 ng of IL-10 were injected into the inguinal WAT of *Lepr^db/db^* mice. The expression levels of IL-6, IL-33, Foxp3, CCL2, IL-1β, TNF-α, IL-10, and iNOS mRNA in SVFs of adipose tissue from *Lepr^db/db^* mice were determined seven days after injection. We observed that the injection of PBS-treated SVFs into the adipose tissue of *Lepr^db/db^* mice did not significantly change the expression levels of IL-6, IL-33, Foxp3, CCL2, IL-1β, TNF-α, IL-10, and iNOS mRNA in SVFs of adipose tissue from *Lepr^db/db^* mice compared with those in the PBS treatment group. However, injection of 10 ng of IL-10-treated SVFs significantly decreased mRNA expression levels of IL-6, IL-33, CCL2, and IL-1β and increased Foxp3 and IL-10 mRNA expression in adipose tissue compared with that in the PBS-treated SVF group. Injection of 100 ng of IL-10-treated SVFs significantly decreased expression levels of IL-6, IL-33, IL-1β, and CCL2 and increased Foxp3 and IL-10 mRNA expression of SVFs from adipose tissue compared with those in the PBS-treated SVF group (Figure 3). Altogether, these results suggest that IL-10-treated SVFs are capable of reducing IL-6, IL-33, CCL2, and IL-1β gene expression and enhancing Foxp3 and IL-10 expression in adipose SVFs of diabetic mice.

### 2.5. Injection of IL-10-Treated SVFs Increases CD4^+^ Tregs in the Adipose Tissue

Induction of IL-10 promotes Treg differentiation [14]. Therefore, we examined whether IL-10-treated SVFs with upregulated Foxp3 expression could be used to increase CD4^+^ Tregs in the adipose tissue of diabetic mice. We conducted flow cytometry analysis to assess the numbers of CD8, CD11b, and PDGFRα cells in the SVFs from adipose tissue. There were no significant differences in CD11b, CD8, and PDGFRα cells between the PBS and IL-10 treatment groups (Figure 4A). This indicated that IL-10 treatment did not alter the components of SVFs. Tregs are characterized by their expression of Foxp3 in CD4^+^ cells. To further examine whether increased Foxp3 expression in *Lepr^db/db^* mice injected with IL-10-treated SVFs influenced CD4^+^ Tregs in the adipose tissue, we conducted a flow cytometry analysis to assess the number of CD4^+^ Foxp3^+^ cells in the adipose tissue. We found that the number of CD4^+^ Foxp3^+^ cells was significantly higher in *Lepr^db/db^* mice injected with 10 or 100 ng IL-10-treated SVFs than in those injected with PBS or PBS-treated SVFs (Figure 4B,C). Collectively, these results suggest that injection of IL-10-treated SVFs increases CD4^+^ Tregs in SVFs from the adipose tissue of diabetic mice.

### 2.6. Injection of IL-10-Treated SVFs Increases Adipose Tissue IL-10 and Blood Leptin Levels and Decreases Plasma Adiponectin Levels and DPP4 Activity

We wanted to examine whether the injection of IL-10-treated SVFs affected the levels of IL-10 and leptin in the adipose tissue, adiponectin levels, and DPP4 activity in the plasma of diabetic mice. Thus, 10 ng or 100 ng of IL-10-treated SVFs were injected into the inguinal WAT of *Lepr^db/db^* mice, and IL-10 and leptin levels in the adipose tissue and adiponectin levels and DPP4 activity in the plasma were measured. Our results demonstrated that an injection of 10 ng of IL-10-treated SVFs significantly increased IL-10 levels in the adipose tissue of *Lepr^db/db^* mice compared to those receiving an injection of PBS (355.5 ± 82.11 vs. 148.6 ± 10.25 pg/mL) (Figure 5A). We investigated whether plasma DPP4 activity was affected by the injection of IL-10-treated SVFs. Notably, under SVF treatment conditions, the injection of SVFs treated with 10 or 100 ng of IL-10 into the adipose tissue of *Lepr^db/db^* mice suppressed plasma DPP4 activity compared with those treated with PBS (162.1 ± 4.22 and 170.1 ± 10.66 vs. 202.3 ± 7.82 pmol/min/mL × 10^−3^, respectively) (Figure 5B). Next, we examined the plasma adiponectin levels in the different groups. Our results demonstrated that the injection of 10 or 100 ng of IL-10-treated SVFs into the adipose tissue of *Lepr^db/db^* mice decreased plasma adiponectin levels compared with the injection of PBS in *Lepr^db/db^* mice (61.01 ± 5.89 and 61.1 ± 4.68 vs. 91.47 ± 5.56 pg/mL) (Figure 5C). We examined the plasma leptin levels in the different groups. Our results demonstrated that the injection of 100 ng of IL-10-treated SVFs into the adipose tissue of *Lepr^db/db^* mice increased plasma leptin levels compared to the injection of PBS-treated SVFs in *Lepr^db/db^* mice (82.45 ± 1.58 vs. 50.39 ± 1.36 ng/mL) (Figure 5D). Altogether, these results demonstrate that the injection of IL-10-treated SVFs induces adipose IL-10 and plasma leptin levels and suppresses plasma adiponectin levels and DPP4 activity in diabetic mice.

### 2.7. Injection of IL-10-Treated SVFs Decreases the Expression of G6PC and PCK1 in the Liver

We examined whether the injection of IL-10-treated SVFs modulated the expression of gluconeogenesis-related genes in the livers of diabetic mice; 10 ng or 100 ng IL-10-treated SVFs were injected into the inguinal WAT of *Lepr^db/db^* mice, and the livers were harvested and subjected to qPCR analysis to determine the expression of proinflammatory cytokines, G6PC, and PCK1 at 7 days post-injection. Our results showed a significant reduction in G6PC and PCK1 mRNA expression in the livers of *Lepr^db/db^* mice injected with 10 or 100 ng IL-10-treated SVFs compared to those injected with PBS-treated SVFs (Figure 6). The injection of 10 or 100 ng of IL-10-treated SVFs into the inguinal WAT of *Lepr^db/db^* mice did not change ICAM, FGF-21, IL-1β, TNF-α, DPP4, or iNOS mRNA expression compared to those receiving the injection of PBS-treated SVFs (Figure 6). The injection of PBS-treated SVFs into the inguinal WAT of *Lepr^db/db^* mice did not change G6PC, PCK1, ICAM, FGF-21, IL-1β, TNF-α, DPP4, and iNOS mRNA expression of livers compared to those receiving an injection of PBS (Figure 6). These results suggest that injection of IL-10-treated SVFs significantly decreases G6PC and PCK1 mRNA without changing ICAM, FGF-21, IL-1β, TNF-α, DPP4, or iNOS mRNA expression of the livers of diabetic mice.

### 2.8. Injection of IL-10-Treated SVFs Increases Akt and STAT3 Protein Activation and Reverses Glucose Intolerance

To investigate whether the injection of IL-10-treated SVFs attenuates insulin resistance in diabetic mice, *Lepr^db/db^* mice were injected with PBS or IL-10-treated SVFs followed by insulin administration, and the protein expression of Akt phosphorylation, ERK phosphorylation, and STAT3 phosphorylation in the livers of *Lepr^db/db^* mice was measured. We observed a significant reduction in Akt phosphorylation in the livers of *Lepr^db/db^* mice compared to that in *Lepr^+/+^* mice following insulin administration (Figure 7A,B), indicating a poor response to insulin treatment in *Lepr^db/db^* mice. However, injection of IL-10-treated SVFs induced Akt phosphorylation in the livers of *Lepr^db/db^* mice following insulin administration (Figure 7A,B). Injection of IL-10-treated SVFs induced STAT3 phosphorylation in the livers of *Lepr^db/db^* mice following insulin administration (Figure 7A and Appendix A). To further examine the effect of IL-10-treated SVFs on diabetes-induced glucose intolerance, a glucose tolerance test was performed in *Lepr^+/+^* and *Lepr^db/db^* mice and in *Lepr^db/db^* mice injected with IL-10-treated SVFs. After fasting, *Lepr^+/+^* mice showed increased blood glucose levels after glucose administration, which returned to normal 1 h after glucose administration. However, *Lepr^db/db^* mice exhibited significantly higher blood glucose levels throughout the first 75 min after glucose administration than *Lepr^+/+^* mice (Figure 7C). Notably, *Lepr^db/db^* mice injected with 10 or 100 ng of IL-10-treated SVFs displayed a significant reduction in blood glucose levels compared to *Lepr^db/db^* mice (Figure 7C). Taken together, these results suggest that the injection of IL-10-treated SVFs enhances Akt and STAT3 activation and attenuates insulin resistance in *Lepr^db/db^* mice.

## 3. Discussion

Stromal stem cells (i.e., MSCs) are currently the most widely used stem cell types in clinical trials. Although stromal stem cells are emerging as the most promising source for allogenic cell therapy, the therapeutic use of MSCs in clinical trials of DM is highly controversial [12,13]. The presence of adipose-derived stem cells within aspirated human fat has been consistently demonstrated using cell culturing and flow cytometry, and the application of such therapies has produced promising results in reconstructive surgery [15,16]. However, the U.S. Food and Drug Administration does not allow the removal of patient cells for implantation during a secondary procedure because of concerns regarding contamination during cell culturing [17]. To increase clinical applicability, adipose tissue-derived stromal cell therapies may demonstrate safe and effective benefits that avoid the need for cell culturing. To the best of our knowledge, the present study is the first in-depth analysis of the effects and mechanism of modulated adipose tissue-derived stromal cells on metabolic regulation in a type 2 DM (T2DM) mouse model. This study paves the way for the development of a new line of treatment for T2DM.

Endogenous IL-10 protects against diet-induced insulin resistance in the liver [7]. Despite the immunosuppressive effects of IL-10, its role in modulating disease progression in DM remains controversial.

Our data clearly demonstrated that IL-10-treated adipose stromal cells could be used to alleviate liver glucose gluconeogenesis, insulin resistance, and DPP4 activity in a mouse model of T2DM. First, our data demonstrated that diabetic SVFs exhibited a significant increase in IL-6 mRNA and protein expression compared to SVFs from non-diabetic mice. IL-10 treatment of SVFs from the adipose tissue of diabetic mice suppressed IL-6, IL-33, CCL2, TNF-α, and IL-1β cytokines but enhanced Foxp3 mRNA expression. Moreover, IL-10 treatment reduced the IL-6, p-mTOR, pJNK, and pNF-κB protein expression of diabetic SVFs. Altogether, our data demonstrated that diabetic SVFs exhibited increased expression of inflammatory cytokines compared to non-diabetic SVFs. IL-10 was capable of suppressing inflammatory cytokines as well as IL-6, p-mTOR, pJNK, and pNF-κB protein expression but increased Foxp3 expression in adipose SVFs in vitro. We further showed that IL-10 treatment repressed the CCL2 and PDGFRα expression of ATMs and the IL-6 expression of NATMs but increased Foxp3 and IL-10 expression of ATMs in *Lepr^db/db^* mice. Next, we injected IL-10-modulated stromal vascular cells into the adipose tissue of diabetic mice. Our data demonstrated that IL-10-treated SVFs were capable of reducing IL-6, IL-33, CCL2, and IL-1β gene expression and enhancing Foxp3 and IL-10 expression in adipose SVFs of *Lepr^db/db^* mice. This suggests that IL-10-treated SVFs can be used to reduce inflammatory cytokines and increase Foxp3 expression in adipose tissue in a mouse model of T2DM. Moreover, the injection of IL-10-treated SVFs increased CD4^+^ Tregs of SVFs from the adipose tissue of diabetic mice and decreased G6PC and PCK1 mRNA without changing ICAM, FGF-21, IL-1β, TNF-α, DPP4, or iNOS mRNA expression of the liver. Notably, injection of IL-10-treated SVFs induced Akt phosphorylation in the livers of *Lepr^db/db^* mice following insulin administration and reduced glucose intolerance. Collectively, we identified that the injection of IL-10-treated SVFs is capable of mitigating diabetes-induced adipose inflammation, liver gluconeogenesis, and insulin resistance, which may be mediated through an increase in Tregs in the adipose tissue. Our data clearly demonstrate that IL-10-treated adipose stromal vascular cells could be used as a novel cell therapy strategy to alleviate liver glucose gluconeogenesis and insulin resistance in T2DM.

IL-10 improves beta cell function, inhibits insulitis progression, and prevents diabetes in animal models [18]. IL-10 inhibits proinflammatory cytokine production and function in vitro and in vivo [19]. However, IL-10 has a short half-life and short range of activity. The mean half-life of recombinant human IL-10 is 2.7 to 4.5 h [20]. The therapeutic value of IL-10 in the treatment of diabetes-related complications remains unclear and requires further investigation. A previous study demonstrated that IL-10 signaling is necessary for the normal suppression of gluconeogenic gene expression, and physiological concentrations of IL-10 and insulin can suppress glucose production in primary hepatocytes [8]. STAT3 phosphorylation is suppressed in the livers of IL-10 receptor beta-deficient mice. This suggests that STAT3 phosphorylation supports IL-10 signaling through the IL-10 receptor. Suppression of STAT3 phosphorylation and G6PC and PCK1 expression in the livers of IL-10 receptor beta-deficient mice was also significantly decreased, indicating that IL-10 signaling, in cooperation with insulin, mediates the suppression of gluconeogenic genes after feeding [8]. Our data further demonstrated that the injection of IL-10-treated SVFs into the adipose tissue of diabetic mice significantly reduced G6PC and PCK1 expression. Moreover, the injection of IL-10-treated SVFs into the adipose tissue of diabetic mice markedly increased STAT3, ERK, and Akt phosphorylation in the liver after insulin injection. This suggests that the injection of IL-10-treated SVFs into the adipose tissue of diabetic mice suppresses gluconeogenic gene expression and insulin resistance through IL-10/STAT3 signaling in the liver. Modulation of cellular phenotypes within the adipose tissue provides a potential means of therapeutic intervention for diabetes. Our data clearly demonstrated that IL-10 treatment modulated the SVFs of the adipose tissue and IL-10-treated SVFs reduced DM-induced insulin resistance and liver gluconeogenesis by enhancing Tregs in the adipose tissue. Previously, adipose-derived stem cells have been shown to decrease liver injury by inhibiting mitochondrial stress (apoptosis and dysfunctional energy metabolism) and alleviating inflammation in a T1DM mouse model [21]. However, our data demonstrated that the injection of PBS-treated SVFs into the adipose tissue of *Lepr^db/db^* mice did not significantly change the expression of liver G6PC and PCK1, adipose IL-10 levels, plasma adiponectin levels, or plasma DPP4 activity. Interestingly, the injection of IL-10-treated SVFs significantly decreased liver G6PC and PCK1 levels, plasma adiponectin levels, and plasma DPP4 activity compared to those in the PBS group. Moreover, the injection of PBS-treated SVFs into the adipose tissue of *Lepr^db/db^* mice did not change the number of Tregs, whereas the injection of IL-10-treated SVFs significantly increased the number of Tregs in SVFs from the adipose tissue of diabetic mice. Altogether, our data suggest that the injection of SVFs alone did not affect diabetes-induced insulin resistance and that IL-10 could modulate SVFs and exert an inhibitory effect on insulin resistance and liver gluconeogenesis by decreasing inflammation and increasing Treg cells in adipose tissue as well as STAT3 activation in the livers of diabetic mice.

Obesity and its associated conditions (e.g., metabolic syndrome, hypertension, and dyslipidemia) are strongly associated with an increase in insulin resistance and the incidence of T2DM as well as cardiovascular disease [22]. Tregs play a critical role in modulating the inflammatory tone of adipose tissue and maintaining insulin sensitivity [2]. The population of Tregs in epididymal fat is markedly decreased in obese animals, and this reduction is strongly associated with insulin resistance [3]. Tregs exert effects on the inhibition of effector T cell activation and proliferation, and modulate innate immune system activities. Moreover, the expansion of Tregs by IL-2 injection decreased adipose tissue inflammation and improved insulin sensitivity through IL-10-mediated suppression of conventional T cell proliferation in mice [23]. Interestingly, although our data showed that treatment with adipose stromal cells did not alter the level of Foxp3 expression in *Lepr^db/db^* SVFs compared to treatment with PBS, the injection of IL-10-treated SVFs significantly increased Foxp3 expression in the adipose tissue of diabetic mice compared to those treated with PBS in vivo. These results suggest that adipose stromal cells do not directly induce Tregs in SVFs. Instead, IL-10-treated adipose stromal cells induce IL-10 expression in SVFs, which subsequently promote Treg expansion in the adipose tissue of diabetic mice. These findings corroborate that IL-10-treated SVFs can be developed as a potential cell therapeutic approach for inducing Tregs in adipose tissue and improving insulin sensitivity in diabetes.

DPP4 activity is correlated with the severity of obesity and diabetes [24], and plasma DPP4 activity is elevated in patients with T2DM and obesity [25]. Similarly, we observed that diabetes-enhanced DPP4 activity in the plasma was closely related to diabetes-induced adipose inflammation and glucose intolerance. Moreover, the injection of IL-10-treated SVFs into adipose tissue significantly decreased plasma DPP-4 activity. Adiponectin is a circulating hormone secreted by adipose tissue that exerts protective effects against metabolic syndrome [26], inflammation, and atherosclerosis [27,28]. Unlike adipokines such as leptin, plasma adiponectin levels are negatively associated with adiposity and decreased in T2DM, obesity, and insulin resistance [29,30]. Our data demonstrate that IL-10-treated adipose stromal cells induced IL-10 expression in adipose tissue, which may subsequently promote Treg expansion in adipose tissue and decrease adiponectin production and plasma DPP4 activity in diabetic mice. Previously, we demonstrated that non-diabetic plasma inhibits M1 but increases M2 cytokine expression in the adipose tissue of diabetic mice. Moreover, non-diabetic plasma-treated SVFs are capable of mitigating diabetes-induced plasma DPP4 activity, liver inflammation, and insulin resistance, which may be mediated by suppressing M1 cytokines and increasing IL-10 and Tregs in the adipose tissue [31]. Here, we further demonstrated that the injection of IL-10-treated SVFs into the adipose tissue of *Lepr^db/db^* mice decreased proinflammatory cytokine expression, suppressed plasma DPP4 activity, and ameliorated insulin resistance. In addition, we found that the injection of IL-10-treated SVFs decreased blood adiponectin levels and increased leptin levels in diabetic mice.

Our study has several limitations. We did not compare the effects of injecting IL-10-treated SVFs with those of injecting IL-10 alone in terms of liver gluconeogenesis and insulin resistance. We did not use IL-10 receptor-deficient mice to elucidate the role of IL-10 in Treg stimulation or STAT3 phosphorylation. Further studies are required to overcome these limitations.

In summary, we demonstrate the potential protective effects of IL-10-modulated adipose-derived stromal cells in decreasing diabetes-induced DPP4 activity, insulin resistance, and liver gluconeogenesis (Figure 8). We found that IL-10 treatment suppressed inflammatory cytokine expression and IL-6, p-mTOR, pJNK, and pNF-κB protein expression but enhanced the Foxp3 mRNA expression of adipose SVFs from diabetic mice. The injection of IL-10-treated SVFs decreased inflammatory cytokines but increased CD4^+^ Tregs of SVFs and IL-10 levels in adipose tissue; suppressed insulin resistance, hepatic G6PC and PCK1 mRNA expression, plasma adiponectin levels, and DPP4 activity; and increased Akt and STAT3 phosphorylation in the livers of diabetic mice. Our results suggest that IL-10-treated adipose stromal vascular cells could be a promising therapeutic strategy for treating diabetes mellitus.

## 4. Materials and Methods

### 4.1. Mice

*Lepr^db/+^* mice were purchased from the Jackson Laboratory (Bar Harbor, ME, USA) and bred to obtain diabetic *Lepr^db/db^* and non-diabetic *Lepr*^+/+^ mice. *Lepr^db/db^* mice with a mutation in the gene encoding the leptin receptor become obese at 3 to 4 weeks of age, and exhibit elevated plasma insulin and blood sugar at 4 to 8 weeks of age. *Lepr*^+/+^ and *Lepr^db/db^* mice were housed in a pathogen-free facility in micro-isolator cages at the Animal Center of Kaohsiung Veterans General Hospital. They were fed with a standard laboratory diet (Altromin 1324, Altromin GmbH, Lage, Germany; 11.9 kJ/g, 19% crude protein, 4% crude fat, 6% crude fiber) and had ad libitum access to food and water. They were housed in individually ventilated cage (IVC) systems with ambient lighting control to provide 12 h light/12 h dark cycles. Both male and female mice were used for all experiments. All animal experimental procedures were designed, performed, and approved by the Institutional Animal Care and Use Committee (IACUC) at Kaohsiung Veterans General Hospital.

### 4.2. Preparation of Stromal Vascular Fractions (SVFs)

Vascular adipose tissue isolated from bilateral inguinal adipose tissue of *Lepr^db/db^* and *Lepr^+/+^* mice at 10 to 12 weeks of age was minced into small pieces and digested with collagenase VIII (Cat# C2139, Sigma-Aldrich, St. Louis, MO, USA) in ice-cold HBSS (2 mg/mL) for 15 min at 37 °C. Cells were then passed through 100 μm cell strainers and centrifuged at 1200 rpm for 10 min. The cell pellets were collected and retrieved as SVFs for experiments. The number of SVF cells was counted with a Cellometer (Nexcelom Bioscience, Lawrence, MA, USA). We harvested 2.1 to 2.6 g of adipose tissue from each *Lepr^db/db^* mouse with 42 g of body weight.

#### In Vitro Treatment of SVFs

For in vitro treatment, 10 ng or 100 ng IL-10 was added to 1 mL of PBS with SVFs (2 × 10^7^ cells), and the mixture was then incubated for 3.5 h at 37 °C [32]. Samples were centrifuged at 1700 rpm for 10 min and washed with phosphate-buffered saline (PBS). The pellets were harvested after centrifugation and subjected to analysis.

### 4.3. In Vivo Injection of IL-10-Treated Adipose Stromal Cells into Adipose Tissue

*Lepr^+/+^* mice received 1 mL PBS injection into adipose tissue over the bilateral inguinal area. *Lepr^db/db^* mice were randomly divided into four groups. Group I, used as controls, received PBS injection into the adipose tissue of the bilateral inguinal area. Group II received the injection of PBS-treated *Lepr^db/db^* SVFs (2 × 10^7^ cells) into the adipose tissue of the bilateral inguinal area. Group III and group IV received an injection of 10 ng or 100 ng IL-10-treated *Lepr^db/db^* SVFs (2 × 10^7^ cells) into the adipose tissue of the bilateral inguinal area. On day 7 post-injection, the animals were sacrificed, and the liver, adipose tissue, and blood were harvested for analysis.

### 4.4. Isolation of ATMs from SVFs with Microbeads

The SVFs were pelleted by centrifugation, washed once with buffer, and incubated with anti-F4/80 microbeads (130-110-443, Miltenyi Biotec, San Diego, CA, USA). The cells were rinsed once with PBS–bovine serum albumin (BSA), pelleted by centrifugation, and resuspended in PBS–BSA. ATMs were isolated using magnetic separation columns, and non-ATMs (NATMs) were collected from the washing solution.

### 4.5. RNA Isolation and Quantitative Real-Time Polymerase Chain Reaction (qPCR)

Total RNA was purified from samples using total RNA Miniprep Purification Kits (GeneMark, GMbiolab, Taichung, Taiwan)) and then reverse-transcribed into cDNA using RT kits (Invitrogen, Carlsbad, CA, USA, Lot# 2234812). For the qPCR assay, 2 µL of 200 ng cDNA was added into a mixture containing 12.5 μL of 2× Fast SYBR Green Master Mix (Applied Biosystems, Cat# 4385612, Waltham, MA, USA), 2.5 μL of sense and anti-sense primers (25 μM), and 8 μL of sterile water to reach the final volume of 25 µL. The amplification was performed by using a StepOnePlus™ Real-Time PCR System (Applied Biosystems 7300).

### 4.6. Western Immunoblots

The expressions of pAkt (Cell Signaling, # 4060), Akt (Cell Signaling, # 4691, Danvers, MA, USA), NF-κB (Cell Signaling, # 8242), pNF-κB p65 (Cell Signaling, # 3033), JNK (Cell Signaling, # 9252), pJNK (Cell Signaling, # 9251), pERK (Cell Signaling, # 9101), STAT3, pSTAT3, and pmTOR were detected by Western immunoblot analysis. Briefly, the harvested tissues were homogenized in protein extraction buffer (Sigma, St. Louis, MO, USA), containing a proteinase inhibitor cocktail (Roche, Basel, Switzerland), and then subjected to SDS-PAGE at 50 to 100 V for 2 h. Following being transferred onto a nitrocellulose membrane, the membrane was blocked with 5% non-fat milk in TBST buffer (10 mM Tris-HCl, pH 7.5, 150 mM NaCl, and 1.2% Tween 20) for 1 h and incubated with the specific primary antibody at room temperature for 1 h. After washing with TBST buffer, the membrane was incubated with the secondary antibody. Following the wash, the protein bands were identified by enhanced chemiluminescence (ECL) detection reagent (Millipore, Burlington, MA, USA).

### 4.7. Flow Cytometry Analysis

Cells in the SVFs were suspended in the staining buffer (PBS containing 0.5% BSA and 2 Mm ethylenediaminetetraacetic acid) and then incubated with CD11b (BioLegend, San Diego, CA, USA, clone: M1/70), CD8a (BioLegend, clone: 53–6.7), and PDGFRα (BioLegend, clone: APA5) antibodies or the control isotypes at 4 °C. Thirty min later, cells were then washed twice and resuspended in the staining buffer. In another experiment, after incubation with 7-amino-actinomycin D (BioLegend), the cells were analyzed by an Attune NxT Flow Cytometer (Thermo Fisher, Waltham, MA, USA). The data analysis was performed by using FlowJo v9 (Tree Star, Ashland, OR, USA). For intracellular Foxp3 staining, PE/Cyanine7 anti-mouse CD4 (BioLegend, # 100528)-stained cells were resuspended in 1 mL of True-Nuclear™ 1× Fix Concentrate in each tube, vortexed, and incubated at room temperature in the dark for 60 min. Then, we added 2 mL of True-Nuclear™ 1× Perm Buffer (BioLegend, #424401) to each tube and then centrifuged the tubes at 2100 rpm at room temperature for 8 min, and discarded the supernatant. Tubes were decanted, blotted, and washed twice with True-Nuclear™ 1× Perm Buffer. We resuspended the cell pellet in 100 µL of the True-Nuclear™ 1× Perm Buffer. For detection of intracellular antigen, we added 1 μL of PE–anti-mouse Foxp3 antibody (BioLegend, # 126404) to each tube and incubated them in the dark at room temperature for 60 min.

### 4.8. Insulin Treatment

For insulin injection, we injected the mice with either phosphate-buffered saline or insulin at 1.25 mU/g body weight and waited for 20 min before sacrificing the mice and harvesting the livers.

### 4.9. Plasma DPP4 Activity

Plasma was harvested from cardiac blood and stored at −20 °C until it was assayed. The activity of plasma DPP4 was measured by a DPP4 assay kit (BioVision, Milpitas, CA, USA, # K779-100). DPP4 cleaves substrates, which results in the releasing of the quenched fluorescent group, AMC (7-Amino-4-Methyl Coumarin), which can then be detected at Ex/Em = 360/460 nm by using a fluorescence reader.

### 4.10. Enzyme-Linked Immunosorbent Assay (ELISA)

Leptin, IL-10, and adiponectin were detected by using mouse ELISA kits from R&D systems (Minneapolis, MN, USA, # DY479-05), ELISA Max (# 431414), and Invitrogen (Waltham, MA, USA, # KMP0041), respectively. Tissues were homogenized in lysis buffer containing 30 mM Tris, pH 7.5, 300 mM NaCl, 2 mM MgCl2, 10% Triton X-100, 2 mM CaCl2, and 20 μg/mL of pepstatin A/leupeptin/aprotinin. The homogenate was centrifuged at 1000× *g*, 4 °C for 15 min and the supernatant was collected for use. The blood was centrifuged at 1000× *g*, 4 °C for 15 min and the serum was then collected for use. The ELISA plates were coated with 100 μL capture antibody at and stored at 4 °C overnight. After washing, 200 μL of assay dilution buffer was added for blocking at room temperature for 1 h. The samples and serial dilutions of standards were added and incubated at 4 °C overnight. After incubating with the detection antibody, avidin-HRP was added and incubated at room temperature for 30 min. The substrate, 3,3′,5,5′-tetramethylbenzidine (TMB), was added and incubated for 15 min. One hundred µL of stop solution was added to stop the reaction, and the plate was then subjected to measurement of absorbance at 450 nm by using an ELISA reader.

### 4.11. Intraperitoneal Glucose Tolerance Tests (IPGTTs)

IPGTTs were conducted to test glucose tolerance. Mice were subjected to a period of 15 h fasting; blood glucose was then measured before and at 15, 30, 45, 60, 75, 90, and 120 min after intraperitoneal injection of glucose (1 g/kg body weight; Sigma, St. Louis, MO, USA). Plasma glucose was measured by using a glucose meter (Accu-Chek Performa; Roche, Basel, Switzerland).

### 4.12. Statistical Analysis

All statistical analyses were performed using GraphPad Prism 8. Data were analyzed by unpaired *t*-test for the comparisons between two groups or by one-way analysis of variance (ANOVA) followed by Tukey’s multiple comparison test for the comparisons between multiple groups. Data are presented as mean ± standard error. Statistical significance was determined as *p* < 0.05.

## Figures and Tables

**Figure 1 ijms-25-08088-f001:**
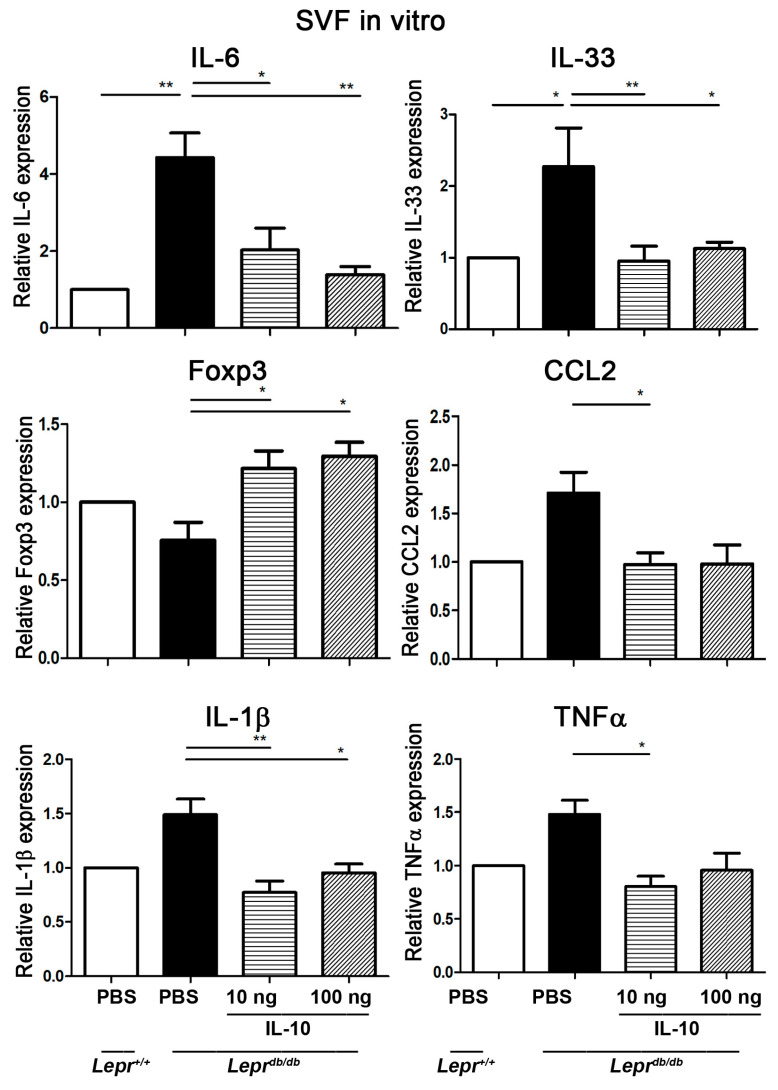
IL-10 suppresses IL-6, IL-33, CCL2, TNF-α, and IL-1β cytokines but enhances Foxp3 mRNA expression in SVFs. SVFs (2 × 10^7^ cells) purified from the adipose tissue of *Lepr^db/db^* mice were treated with PBS, 10 ng IL-10, or 100 ng IL-10 in vitro. Three and a half hours after treatment, cells were collected and subjected to qPCR analysis for determination of IL-6, IL-33, Foxp3, CCL2, IL-1β, and TNF-α cytokine mRNA expression. N = 5/group. * *p* < 0.05, ** *p* < 0.01. Stromal vascular fractions, SVFs; forkhead box p3, *Foxp3*.

**Figure 2 ijms-25-08088-f002:**
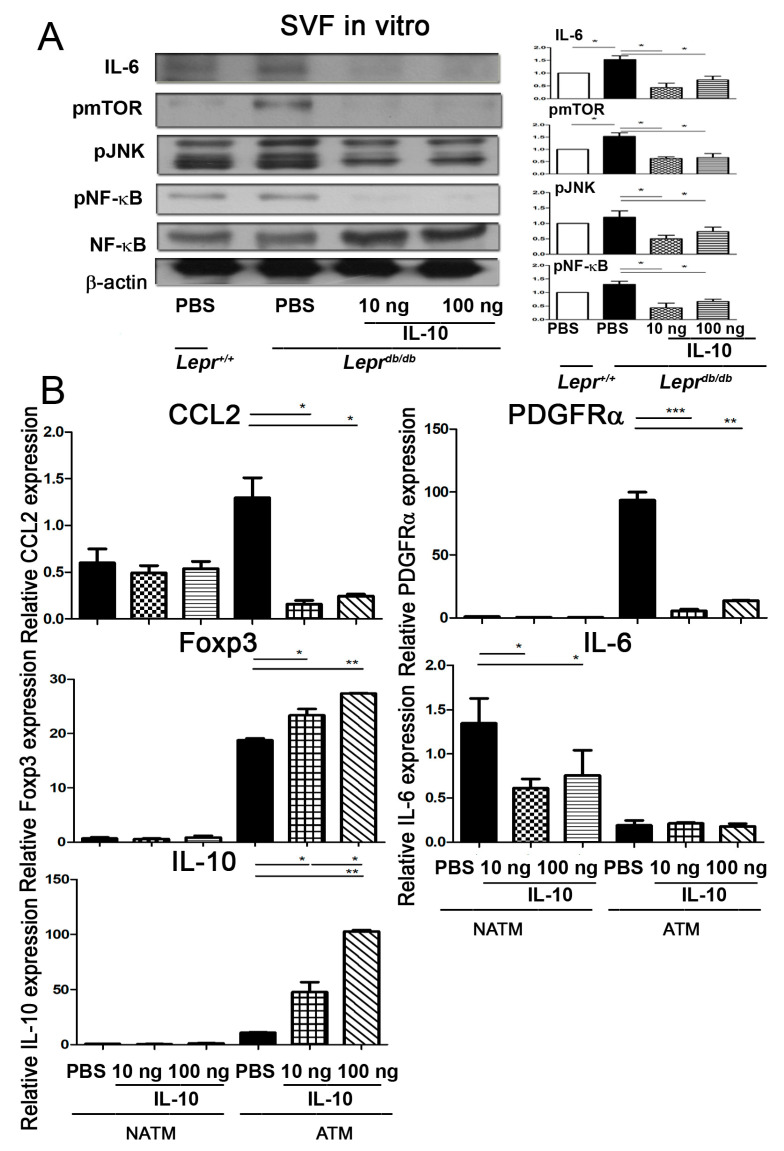
IL-10 suppresses IL-6, pmTOR, pJNK, and pNF-κB protein expression in SVFs and CCL2 and PDGFRα but increases IL-10 and Foxp3 expression in ATMs. (**A**) SVFs were harvested from the adipose tissue of *Lepr^db/db^* mice and treated with PBS, 10 ng IL-10, or 100 ng IL-10 for 3.5 h followed by Western blotting analysis of IL-6, pmTOR, pJNK, and pNF-κB protein expression. Full-length blots/gels are presented in Appendix A. (**B**) To examine whether IL-10 treatment decreases CCL2 and increases Foxp3 as well as IL-10 cytokine expression in ATMs of SVFs from diabetic mice, SVFs harvested from the adipose tissue of *Lepr^db/db^* mice were treated with PBS, 10 ng IL-10, or 100 ng IL-10 for 210 min. ATMs and NATMs were purified from the SVFs followed by qPCR analysis to assess the expression of CCL2, PDGFRα, Foxp3, IL-6, and IL-10 mRNA expression. * *p* < 0.05, ** *p <* 0.01, *** *p* < 0.001. Stromal vascular fractions, SVFs; adipose tissue macrophage, ATM; non-adipose tissue macrophage, NATM; platelet-derived growth factor, *PDGFRα;* forkhead box p3, *Foxp3*.

**Figure 3 ijms-25-08088-f003:**
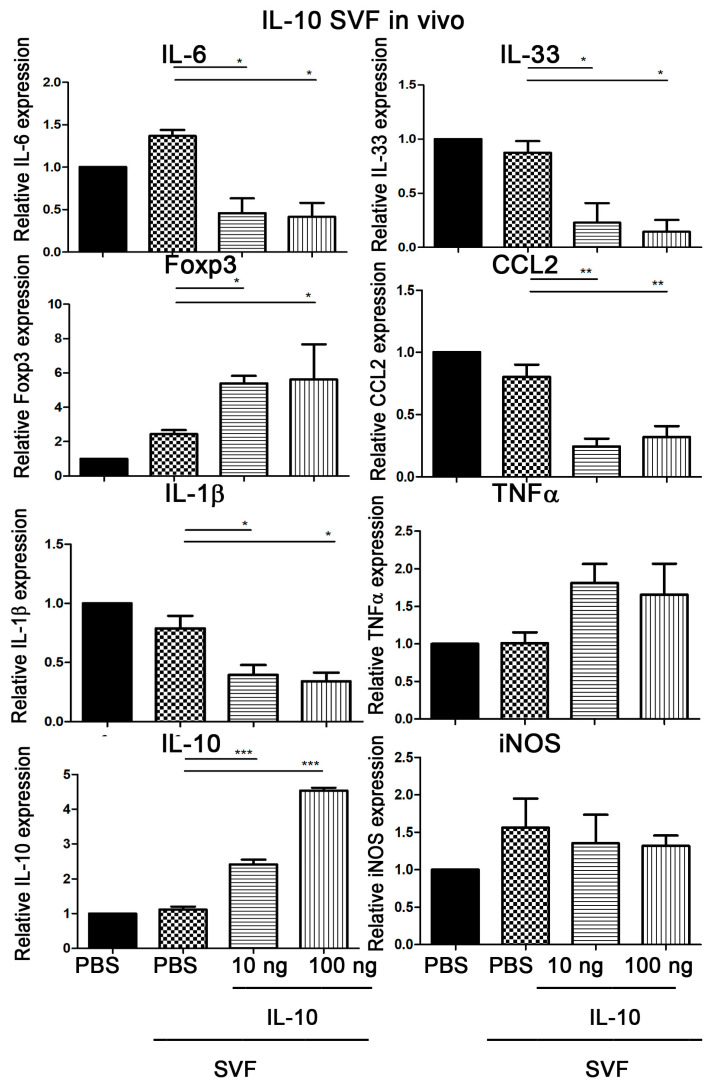
Injection of IL-10-treated SVFs decreases IL-6, IL-33, IL-1β, and CCL2 but enhances Foxp3 and Il-10 expression in the adipose tissue. SVFs were harvested from the adipose tissue of *Lepr^db/db^* mice treated with PBS, 10 ng IL-10, or 100 ng IL-10 for 210 min and injected into the inguinal WAT of *Lepr^db/db^* mice. The expression of IL-6, IL-33, Foxp3, CCL2, IL-1β, TNF-α, IL-10, and iNOS mRNA in SVFs of adipose tissue from *Lepr^db/db^* mice were examined by qPCR analysis seven days after injection. N = 5/group. * *p* < 0.05, ** *p* < 0.01, *** *p* < 0.001. Forkhead box p3, *Foxp3*.

**Figure 4 ijms-25-08088-f004:**
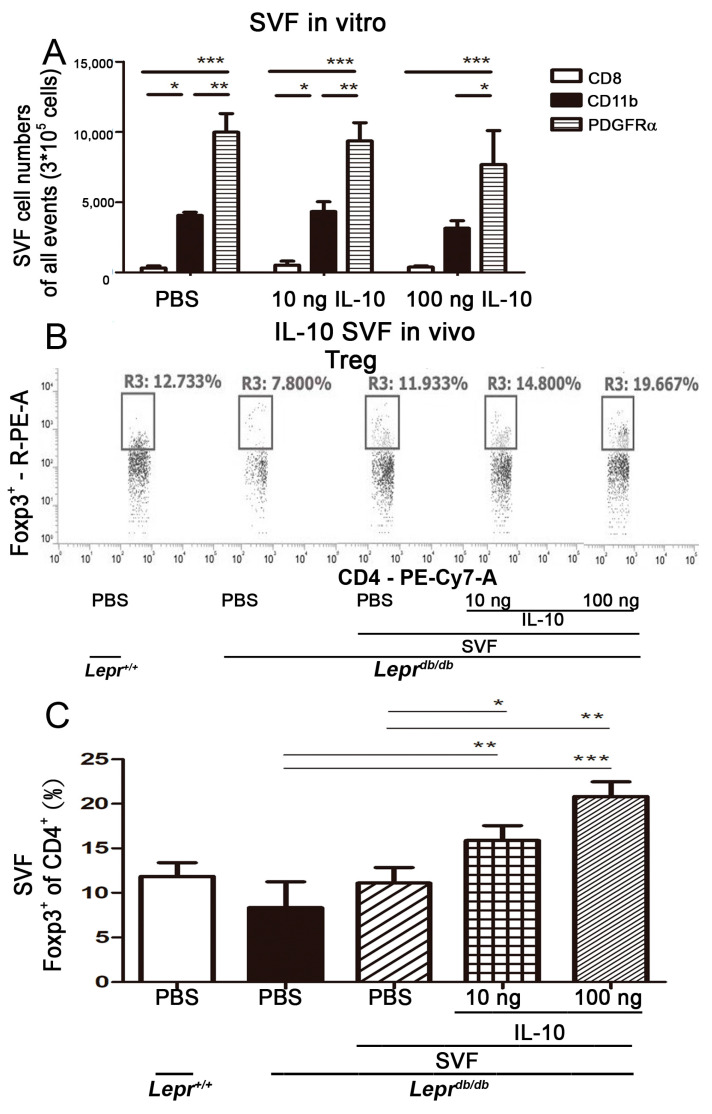
Injection of IL-10-treated SVFs increases CD4^+^ Tregs in the adipose tissue. SVFs (2 × 10^7^ cells) purified from the adipose tissue of *Lepr^db/db^* mice were treated with PBS, 10 ng IL-10, or 100 ng IL-10 in vitro. Three and a half hours after treatment, cells were collected. (**A**) Flow cytometry analysis was used to assess the numbers of CD8, CD11b, and PDGFRα cells in the SVFs. (**B**,**C**) SVFs were harvested from the adipose tissue of *Lepr^db/db^* mice treated with PBS, 10 ng IL-10, or 100 ng IL-10 for 210 min and injected into the inguinal WAT of *Lepr^db/db^* mice. Flow cytometry analysis was used to assess the frequency and number of CD4^+^ Tregs in the adipose tissue. N = 5/group. * *p* < 0.05, ** *p <* 0.01, *** *p* < 0.001.

**Figure 5 ijms-25-08088-f005:**
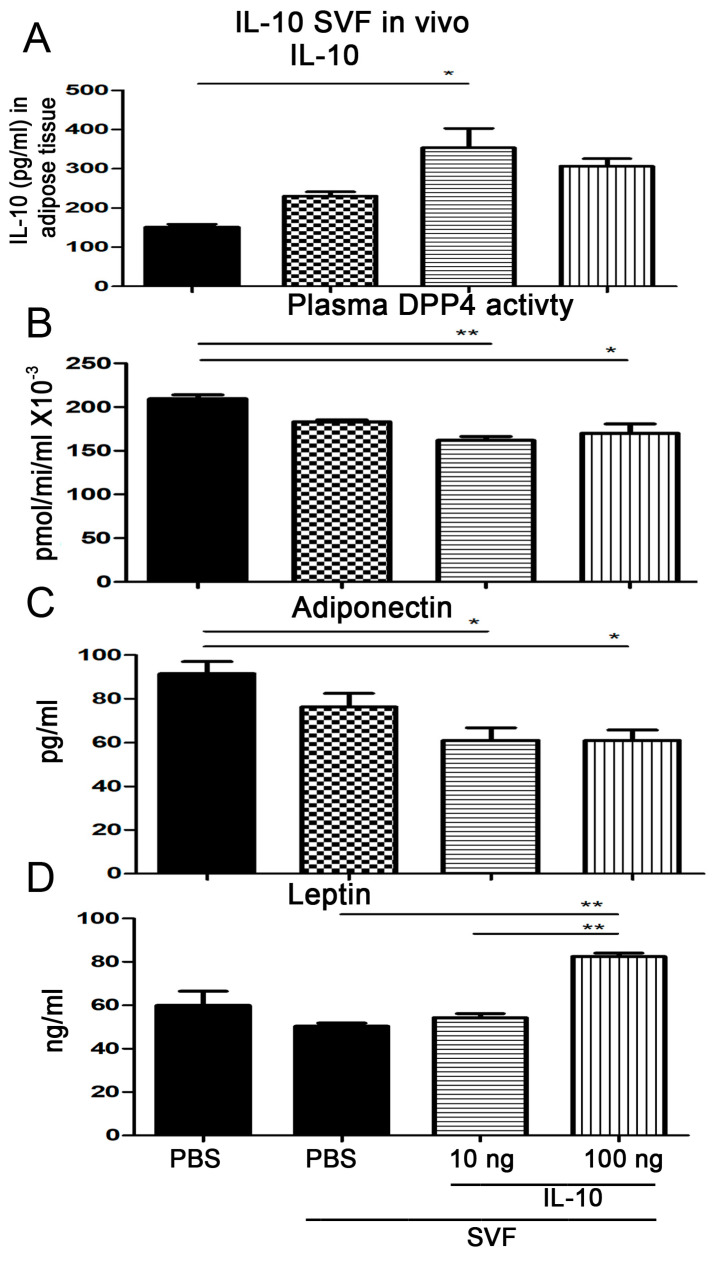
Injection of IL-10-treated SVFs increases adipose tissue IL-10 and leptin levels and decreases plasma adiponectin levels and DPP4 activity in *Lepr^db/db^* mice. SVFs were harvested from the adipose tissue of *Lepr^db/db^* mice and treated with PBS, 10 ng IL-10, or 100 ng IL-10. PBS or IL-10-treated SVFs were injected into the inguinal WAT of *Lepr^db/db^* mice. One week later, the adipose tissue was harvested and then subjected to ELISA to detect IL-10 (**A**) levels. The blood was harvested to measure plasma DPP4 activity (**B**). Subsequently, the blood was harvested and subjected to ELISA to determine adiponectin (**C**) and leptin (**D**) levels. N = 5/group. * *p* < 0.05; ** *p* < 0.01.

**Figure 6 ijms-25-08088-f006:**
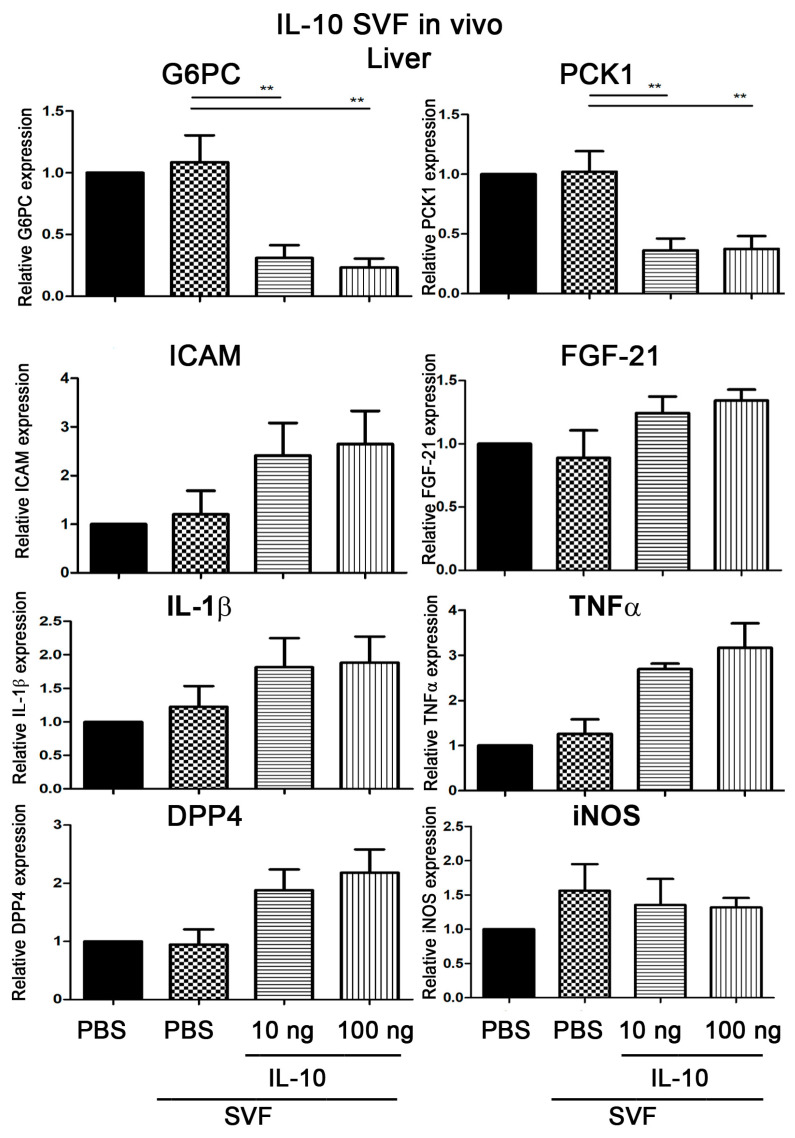
Injection of IL-10-treated SVFs decreases the expression of G6PC and PCK1 in the liver. SVFs were harvested from the adipose tissue of *Lepr^db/db^* mice treated with PBS, 10 ng IL-10, or 100 ng IL-10 for 3.5 h and injected into the inguinal WAT of *Lepr^db/db^* mice. The livers were harvested and subjected to qPCR analysis to determine the G6PC, PCK1, ICAM, FGF-21, IL-1β, TNF-α, DPP4, and iNOS mRNA expression of the liver at 7 days post-injection. N = 6/group. ** *p* < 0.01.

**Figure 7 ijms-25-08088-f007:**
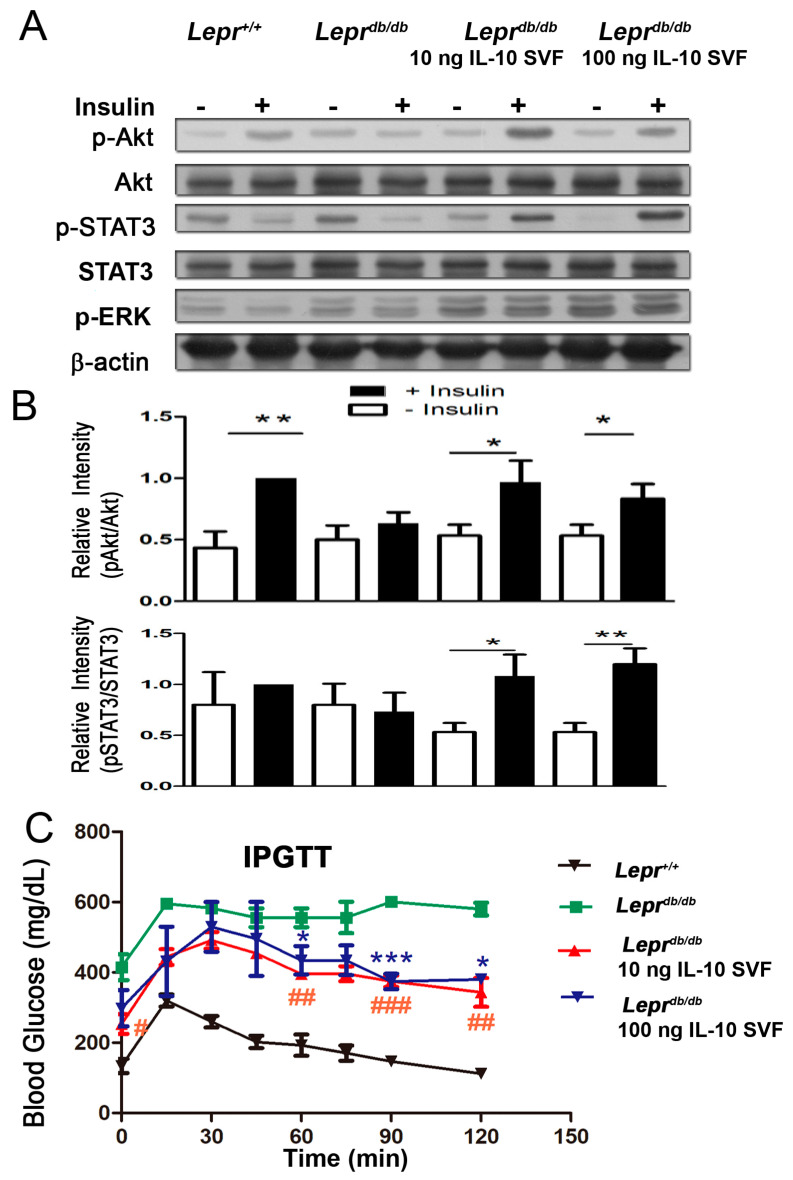
Injection of IL-10-treated SVFs increases Akt activation and attenuates glucose intolerance in *Lepr^db/db^* mice. (**A**) SVFs were harvested from the adipose tissue of *Lepr^db/db^* mice treated with PBS, 10 ng IL-10, or 100 ng IL-10 for 3.5 h and injected into the inguinal WAT of *Lepr^db/db^* mice. One week after injection, mice were treated with vehicle or insulin, 1.25 mIU/g body weight for 20 min. Mice were then sacrificed and the livers were harvested, followed by Western blot analysis for pAkt, Akt, pERK, STAT3, and pSTAT3 expression. Full-length blots/gels are presented in Appendix A. (**B**) The ratios of pAkt/Akt and pSTAT3/STAT3 were also calculated. * *p* < 0.05; ** *p* < 0.01. (**C**) *Lepr^db/db^* mice treated with PBS-treated or IL-10-treated SVFs were administered with glucose (1 g/kg body weight; Sigma). The blood glucose level was measured before and every 15 min up to 120 min after glucose administration by using a glucose meter. N = 4/group. * *p* < 0.05; *** *p* < 0.001 compared to *Lepr^+/+^* mice. **^#^**
*p* < 0.05; **^##^**
*p* < 0.01; **^###^** *p* < 0.001 compared to *Lepr^+/+^* mice.

**Figure 8 ijms-25-08088-f008:**
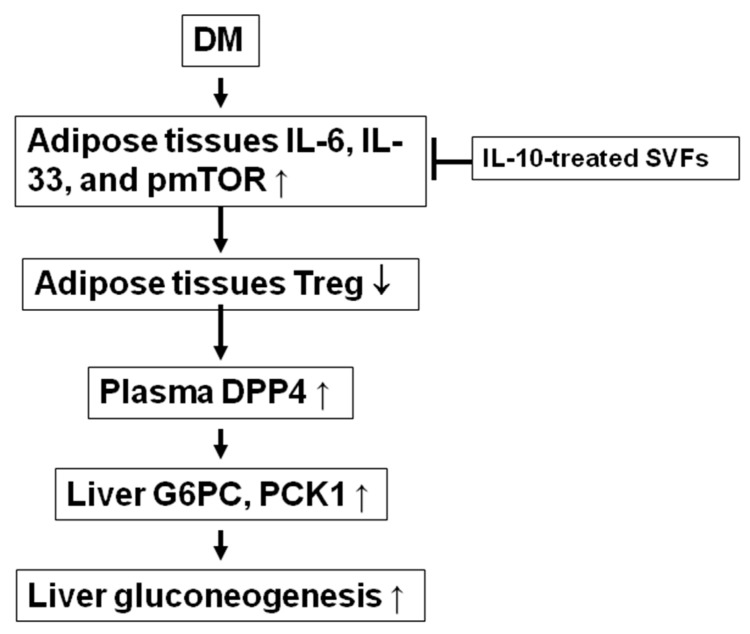
The model of IL-10-treated SVFs enhancing Treg cells and attenuating gluconeogenesis, DPP4 activity, and insulin resistance in diabetic mice. IL-10 treatment suppresses inflammatory cytokine expression and IL-6, p-mTOR, pJNK, and pNF-κB protein expression but enhances Foxp3 mRNA expression of adipose SVFs from diabetic mice. Injection of IL-10-treated SVFs decreases inflammatory cytokines but increases CD4^+^ Tregs of SVFs in adipose tissue; suppresses insulin resistance, hepatic G6PC and PCK1 mRNA expression, and DPP4 activity; and increases Akt and STAT3 phosphorylation in the livers of diabetic mice. Increase,↑; decrease↓.

## Data Availability

All relevant data and material to reproduce the findings are available in the manuscript.

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
