# Peer review of "IL-10 Enhances the Inhibitory Effect of Adipose-Derived Stromal Cells on Insulin Resistance/Liver Gluconeogenesis by Treg Cell Induction"

_ijms, 2024, doi:10.3390/ijms25158088_

Round 1

Reviewer 1 Report

Comments and Suggestions for Authors

See the annex for specific amendments

Comments on the Quality of English Language

English is appropriately expressed.

Author Response

General opinion: This study investigated the role and mechanism of IL-10-treated adipose stromal cells on diabetes induced insulin resistance and liver regeneration, in order to provide a new perspective for the treatment of diabetes. However, the description and analysis of the results here is limited to changes in relevant measures after drug treatment, lacks systematic description, and does not clarify how to exclude the effects of other pathways. In addition, the charts are not clear, the notes on the charts are ambiguous, and the font size is inconsistent throughout the document. Therefore, I would not recommend this paper for publication.

Comments are as follows:

Major comments

  1. In the full description, sometimes it is said that the therapeutic effect of IL-10 on diabetes (Page 15, line 366), and sometimes it is said that IL-10 is effective in the treatment of diabetic complications (Page 1, lines 31 and 32), which is very confusing, please reorganize it in combination with the experimental results and repeatedly confirm the correctness of the expression. In addition, there are many complications of diabetes, and the authors still need to consider whether there is an over-the-top formulation here.

Ans: Thanks for suggestion. Liver gluconeogenesis has been recognized as a complication of DM. We mainly evaluated the effects of IL-10-treated SVFs on liver gluconeogenesis and insulin resistance in a T2DM mouse model. We have revised those different descriptions in this manuscript.

  1. Do not compress the image (Fig2(A)、Fig7(B)), and allocate the ratio of the horizontal and vertical coordinates reasonably to increase the intuitiveness of the article. The group labeling of the pictures in the article is not clear, and the layout is confusing, please rearrange it.

Ans: Thanks for suggestion. Fig 2(A) and Fig7(B) have revised. Group labeling of the picture and layouts have been checked and revised.

  1. This paper only reviews the changes of relevant indicators after IL-10 intervention, and does not systematically sort out the mechanism of action and regulation mode, which lacks logic, and suggests supplementing the summary of the regulatory roadmap.

Ans: Thanks for suggestion. We have found out that injection of IL-10-treated SVFs induced adipose IL-10 and plasma leptin levels and suppressed plasma adiponectin levels and DPP4 activity in diabetic mice (Fig. 5). The summary of the regulatory roadmap has been added in the Fig. 8.

  1. The text directly examines the NF-κB related pathways without explaining how other pathway possibilities are excluded. Please provide additional clarification to make the article more rigorous and accurate.

Ans: IL-10 is an immunoregulatory cytokine that plays an important role in the modulation of inflammation [1]. We have checked the JNK and NF-kB pathways and found that IL-10 treatment suppresses IL-6, p-mTOR, p-JNK, and pNFκB protein expression in diabetic SVFs. (Fig. 2).

Minor comments 1. Please check whether the text of the article needs to be bold according to the format requirements of the magazine, such as lines 111-122 on page 4.

Ans: Thanks for reminding. We have corrected the font and bold of the text in this manuscript.

The first paragraph of the preface seems to introduce the regulatory role of immune cells on inflammation of adipose tissue tissue, but the introduction is very vague, and it is recommended to further focus on the Treg cell explored in this article.

Ans: Thanks for suggestion. We have revised the preface and put Treg cells part in the first paragraph of the preface.

  1. Figure 7(A) lacks a quantification map, please supplement and analyze its significant differences.

Ans: We have added the difference of pSTAT3 between different groups in figure 7B.

  1. The font size in the article is not uniform (14 pages, lines 344-346,page 16, lines 431-439), please refer to the format requirements of the magazine to correct it and check the full text.

Ans: Thanks for suggestion. The font size in page 14 and 16 has been checked and corrected.

  1. Please check that the units used on page 18, line 522 of the article, are correct.

Ans: Thanks for suggestion. The units used on page 18 have been corrected.

This manuscript studied IL10-treated adipose -derived stroll cells from diabetic db/db mice in vitro and in vivo on insulin resistance and liver gluconeogenesis. They found IL-10 treatment suppressed mRNA expression of Inflammation gene  and  enhanced Foxp3 mRNA expression in SVFs from diabetic mice. Injection of IL-10-treated SVFs increased CD4+ regulatory T cells (Tregs) in SVFs in adipose tissue, decreased hepatic G6PC and PCK1 mRNA expression,  and improved  glucose tolerance in diabetic mice in diabetic mice. Their  data suggest that IL-10-treated adipose stromal vascular cells could be a promising therapeutic strategy for treating the complications of diabetes mellitus.  The concept is intriguing. However,the following isuses need to be addressed and discussed to increase the quality of this manuscript.

  1. IL10 injection is an easy experiment, analysis of adipose tissue cell type changes and insulin sensitivity may improve the observations in vivo.

Ans: IL-10 has a short half-life and short range of activity. The mean half-life of recombinant human IL-10 is 2.7 to 4.5 h [2]. We have examined the effects of injection of IL-10 into adipose tissue on insulin sensitivity in a T2DM mouse model and found no significant change after IL-10 injection. Therefore, we used IL-10-treated SVFs and found the significant difference in our model.

  1. IL10 increase macrophage polarization from M1 to M2. M1 and M2 macrophage could be analyzed by FACS.

Ans: SVFs (2 × 107 cells) purified from the adipose tissue of Leprdb/db mice were treated with PBS, 10 ng IL-10, or 100 ng IL-10 in vitro. Three and a half hours after treatment, cells were collected. Flow cytometry analysis was used to assess the numbers of CD8, CD11b, and PDGFRα cells in the SVFs. There were no significant differences in CD11b, CD8, and PDGFRα cells between the PBS and IL-10 treatment groups (Fig. 4A).

  1. The injected SVFs may function directly in inguinal adipose tissues for adipose tissue remodeling. This data was missing.  Adipose tissue morphology and function need to be shown to support the conclusion. Such as adipocytes size in fat tissues (H&E) and insulin signaling  in fat.

Ans:Thanks for suggestion. We had examined the function of adipose tissue in different groups.

Leprdb/db mice were randomly divided into four groups. Group I received PBS injection into the adipose tissue of the bilateral inguinal area as controls. Group II received the injection of PBS-treated Leprdb/db SVFs (2 × 107 cells). Group III and group IV received the injection of 10 ng or 100 ng IL-10-treated Leprdb/db SVFs (2 × 107 cells).

To examine whether increased Foxp3 expression in Leprdb/db mice injected with IL-10-treated SVFs influenced CD4+ Tregs in the adipose tissue, we conducted a flow cytometry analysis to assess the number of CD4+ Foxp3+ cells in the adipose tissue. We found that the number of CD4+ Foxp3+ cells was significantly higher in Leprdb/db mice injected with 10 or 100 ng IL-10-treated SVFs than in those injected with PBS or PBS-treated SVFs (Fig. 4B and C). However, injection of SVFs into adipose tissue did not chagne the number of CD4+ Foxp3+ cells compared with PBS injection group (Fig. 4B and C).

To evaluate the function of adipose tissue, we also examined plasma DPP4 activity, plasma adiponectin, and plasma leptin levels in the different groups. Our results demonstrated that the injection of SVFs into the adipose tissue of Leprdb/db mice did not change plasma DPP4 activity, plasma adiponectin levels, nor plasma leptin levels compared with the injection of PBS in Leprdb/db mice. However, injection of 100 ng of IL-10-treated SVFs into the adipose tissue of Leprdb/db mice increased plasma leptin levels compared to the injection of PBS-treated SVFs in Leprdb/db mice (Fig. 5D). Altogether, these results demonstrate that injection of IL-10-treated SVFs induces adipose IL-10 and plasma leptin levels and suppresses plasma adiponectin levels and DPP4 activity in diabetic mice.

  1. Use IL10 treated NATM or ATM injection may show the effect in adipose tissue remodeling. It is not clear which cell type is the main driving force for the obervasion.

Ans: We have tried to examine the effect of injection of IL10-treated NATM or ATM into adipose tissue in a T2DM mouse model. However, due to the scarce of the ATM in SVFs from adipose tissue, we data did not show any significant result of injection of IL10-treated NATM on insulin resistance or adipose function change in a T2DM mouse model.

  1. Insulin tolerance test or insulin signaling in fat and muscle could support on reduced insulin resistance.

Ans: Thanks for suggestion. We did not examine the insulin reistance in fat or muscle in our mouse model. We will try to examine it in next experiments.

Below errors need to be corrected:

  1. Figure 2A  second PBS pJNK WB signal needs to be improved. I cannot see the higher pJNK compare to the first line PBS control bend. Please label protein marker on the western blot in supplement figures.

Ans: We have revised PBS pJNK western blotting image in figure 2A. We have label protein marker on the western blot in supplement figures.

  1. Fig2B PDGFa and PDGFRa were wrong labeled, please check fig2 legend and page 6  line 138 and 144 and correct them.

Ans: Fig 2B has been corrected.

  1. Figure 5  legend figure C and D miss labeled. Leptin and adiponectin secreted by adipose tissues need to be checked and discussed properly.

Ans: Thanks for correction. We have revised figure 5 legend.

  1. Figure 7C label is wrong, please correct it.

Ans: The label of figure 7C has been corrected.

References:

  1. Couper KN, Blount DG, Riley EM. IL-10: the master regulator of immunity to infection. J Immunol. 2008;180(9):5771-7. Epub 2008/04/22. doi: 10.4049/jimmunol.180.9.5771. PubMed PMID: 18424693.
  2. Saxena A, Khosraviani S, Noel S, Mohan D, Donner T, Hamad AR. Interleukin-10 paradox: A potent immunoregulatory cytokine that has been difficult to harness for immunotherapy. Cytokine. 2015;74(1):27-34. doi: 10.1016/j.cyto.2014.10.031. PubMed PMID: 25481648; PubMed Central PMCID: PMCPMC4454631.

.

Reviewer 2 Report

Comments and Suggestions for Authors

This manuscript studied IL10-treated adipose -derived stroll cells from diabetic db/db mice in vitro and in vivo on insulin resistance and liver gluconeogenesis. They found IL-10 treatment suppressed mRNA expression of Inflammation gene  and  enhanced Foxp3 mRNA expression in SVFs from diabetic mice. Injection of IL-10-treated SVFs increased CD4+ regulatory T cells (Tregs) in SVFs in adipose tissue, decreased hepatic G6PC and PCK1 mRNA expression,  and improved  glucose tolerance in diabetic mice in diabetic mice. Their  data suggest that IL-10-treated adipose stromal vascular cells could be a promising therapeutic strategy for treating the complications of diabetes mellitus.  The concept is intriguing. However,the following isuses need to be addressed and discussed to increase the quality of this manuscript.

1.IL10 injection is an easy experiment, analysis of adipose tissue cell type changes and insulin sensitivity may improve the observations in vivo.

2. IL10 increase macrophage  polarization from M1 to M2. M1 and M2 macrophage could be analyzed by FACS.

3.The injected SVFs may function directly in inguinal adipose tissues for adipose tissue remodeling. This data was missing.  Adipose tissue morphology and function need to be shown to support the conclusion. Such as adipocytes size in fat tissues (H&E) and insulin signaling  in fat.

4. Use IL10 treated NATM or ATM injection may show the effect in adipose tissue remodeling. It is not clear which cell type is the main driving force for the obervasion.

5. Insulin tolerance test or insulin signaling in fat and muscle could support on reduced insulin resistance.

Below errors need to be corrected:

1.Figure 2A  second PBS pJNK WB signal needs to be improved. I cannot see the higher pJNK compare to the first line PBS control bend. Please label protein marker on the western blot in supplement figures.

2. Fig2B PDGFa and PDGFRa were wrong labeled, please check fig2 legend and page 6  line 138 and 144 and correct them.

3. Figure 5  legend figure C and D miss labeled. Leptin and adiponectin secreted by adipose tissues need to be checked and discussed properly.

4. Figure 7C label is wrong, please correct it.

Comments on the Quality of English Language

Good.

Author Response

  1. This manuscript studied IL10-treated adipose -derived stroll cells from diabetic db/db mice in vitro and in vivo on insulin resistance and liver gluconeogenesis. They found IL-10 treatment suppressed mRNA expression of Inflammation gene  and  enhanced Foxp3 mRNA expression in SVFs from diabetic mice. Injection of IL-10-treated SVFs increased CD4+ regulatory T cells (Tregs) in SVFs in adipose tissue, decreased hepatic G6PC and PCK1 mRNA expression,  and improved  glucose tolerance in diabetic mice in diabetic mice. Their  data suggest that IL-10-treated adipose stromal vascular cells could be a promising therapeutic strategy for treating the complications of diabetes mellitus.  The concept is intriguing. However,the following isuses need to be addressed and discussed to increase the quality of this manuscript.

    1. IL10 injection is an easy experiment, analysis of adipose tissue cell type changes and insulin sensitivity may improve the observations in vivo.

    Ans: IL-10 has a short half-life and short range of activity. The mean half-life of recombinant human IL-10 is 2.7 to 4.5 h [2]. We have examined the effects of injection of IL-10 into adipose tissue on insulin sensitivity in a T2DM mouse model and found no significant change after IL-10 injection. Therefore, we used IL-10-treated SVFs and found the significant difference in our model.

    1. IL10 increase macrophage polarization from M1 to M2. M1 and M2 macrophage could be analyzed by FACS.

    Ans: SVFs (2 × 107 cells) purified from the adipose tissue of Leprdb/db mice were treated with PBS, 10 ng IL-10, or 100 ng IL-10 in vitro. Three and a half hours after treatment, cells were collected. Flow cytometry analysis was used to assess the numbers of CD8, CD11b, and PDGFRα cells in the SVFs. There were no significant differences in CD11b, CD8, and PDGFRα cells between the PBS and IL-10 treatment groups (Fig. 4A).

    1. The injected SVFs may function directly in inguinal adipose tissues for adipose tissue remodeling. This data was missing.  Adipose tissue morphology and function need to be shown to support the conclusion. Such as adipocytes size in fat tissues (H&E) and insulin signaling  in fat.

    Ans:Thanks for suggestion. We had examined the function of adipose tissue in different groups.

    Leprdb/db mice were randomly divided into four groups. Group I received PBS injection into the adipose tissue of the bilateral inguinal area as controls. Group II received the injection of PBS-treated Leprdb/db SVFs (2 × 107 cells). Group III and group IV received the injection of 10 ng or 100 ng IL-10-treated Leprdb/db SVFs (2 × 107 cells).

    To examine whether increased Foxp3 expression in Leprdb/db mice injected with IL-10-treated SVFs influenced CD4+ Tregs in the adipose tissue, we conducted a flow cytometry analysis to assess the number of CD4+ Foxp3+ cells in the adipose tissue. We found that the number of CD4+ Foxp3+ cells was significantly higher in Leprdb/db mice injected with 10 or 100 ng IL-10-treated SVFs than in those injected with PBS or PBS-treated SVFs (Fig. 4B and C). However, injection of SVFs into adipose tissue did not chagne the number of CD4+ Foxp3+ cells compared with PBS injection group (Fig. 4B and C).

    To evaluate the function of adipose tissue, we also examined plasma DPP4 activity, plasma adiponectin, and plasma leptin levels in the different groups. Our results demonstrated that the injection of SVFs into the adipose tissue of Leprdb/db mice did not change plasma DPP4 activity, plasma adiponectin levels, nor plasma leptin levels compared with the injection of PBS in Leprdb/db mice. However, injection of 100 ng of IL-10-treated SVFs into the adipose tissue of Leprdb/db mice increased plasma leptin levels compared to the injection of PBS-treated SVFs in Leprdb/db mice (Fig. 5D). Altogether, these results demonstrate that injection of IL-10-treated SVFs induces adipose IL-10 and plasma leptin levels and suppresses plasma adiponectin levels and DPP4 activity in diabetic mice.

    1. Use IL10 treated NATM or ATM injection may show the effect in adipose tissue remodeling. It is not clear which cell type is the main driving force for the obervasion.

    Ans: We have tried to examine the effect of injection of IL10-treated NATM or ATM into adipose tissue in a T2DM mouse model. However, due to the scarce of the ATM in SVFs from adipose tissue, we data did not show any significant result of injection of IL10-treated NATM on insulin resistance or adipose function change in a T2DM mouse model.

    1. Insulin tolerance test or insulin signaling in fat and muscle could support on reduced insulin resistance.

    Ans: Thanks for suggestion. We did not examine the insulin reistance in fat or muscle in our mouse model. We will try to examine it in next experiments.

    Below errors need to be corrected:

    1. Figure 2A  second PBS pJNK WB signal needs to be improved. I cannot see the higher pJNK compare to the first line PBS control bend. Please label protein marker on the western blot in supplement figures.

    Ans: We have revised PBS pJNK western blotting image in figure 2A. We have label protein marker on the western blot in supplement figures.

    1. Fig2B PDGFa and PDGFRa were wrong labeled, please check fig2 legend and page 6  line 138 and 144 and correct them.

    Ans: Fig 2B has been corrected.

    1. Figure 5  legend figure C and D miss labeled. Leptin and adiponectin secreted by adipose tissues need to be checked and discussed properly.

    Ans: Thanks for correction. We have revised figure 5 legend.

    1. Figure 7C label is wrong, please correct it.

    Ans: The label of figure 7C has been corrected.

    References:

    1. Couper KN, Blount DG, Riley EM. IL-10: the master regulator of immunity to infection. J Immunol. 2008;180(9):5771-7. Epub 2008/04/22. doi: 10.4049/jimmunol.180.9.5771. PubMed PMID: 18424693.
    2. Saxena A, Khosraviani S, Noel S, Mohan D, Donner T, Hamad AR. Interleukin-10 paradox: A potent immunoregulatory cytokine that has been difficult to harness for immunotherapy. Cytokine. 2015;74(1):27-34. doi: 10.1016/j.cyto.2014.10.031. PubMed PMID: 25481648; PubMed Central PMCID: PMCPMC4454631.

Round 2

Reviewer 1 Report

Comments and Suggestions for Authors

1.Please summarize and condense all the subheadings in the results in the article, the columns should be detailed, and the subheadings in the results should be more concise.

2.Please be consistent with whether the fonts in columns 1、3 and 4 need to be bold, and check for similar errors in the full text.

3.In Figure 2, "pNFkB" is incorrect.

4.It is recommended to unify the units of time in the article, either in hours or in minutes.

Comments on the Quality of English Language

The English grammar of the article is just right and reasonable, and the sentences are fluent, but there are minor problems such as the lack of spaces between words (such as the third line of the discussion section).

Author Response

Comments and Suggestions for Authors

  1. Please summarize and condense all the subheadings in the results in the article, the columns should be detailed, and the subheadings in the results should be more concise.

Ans: Thanks for suggestion. The subheadings in the results have been revised.

  1. Please be consistent with whether the fonts in columns 1、3 and 4 need to be bold, and check for similar errors in the full text.

Ans: Thank for reminding. The fonts in columns 1, 3, and 4 have been revised.

3.In Figure 2, "pNFkB" is incorrect.

Ans: We have corrected "pNFkB" in figure 2.

4.It is recommended to unify the units of time in the article, either in hours or in minutes.

Ans: We have unified the units of time in this article.

Reviewer 2 Report

Comments and Suggestions for Authors

The revised manuscript has been improved. 

In figure 4A, need to change PDGFa to PDGFRa;

Page 17 line 8-9, CCL2 repeat two times, need to remove one, same in the page 20 fig3 legend.

Need to show inguinal fat HE images in supplement. 

Author Response

In figure 4A, need to change PDGFa to PDGFRa;

Ans: Thanks for correction. Figure 4a has been revised.

Page 17 line 8-9, CCL2 repeat two times, need to remove one, same in the page 20 fig3 legend.

Ans: Thanks for correction. These errors have been corrected.

Need to show inguinal fat HE images in supplement. 

Ans: We have examined the changes of immune cells of adipose tissue in different treatments. However, we did not check the fat HE images. Thanks for suggestion. We will do the HE images in next experiment.